# Are There Any Differences between First Grade Boys and Girls in Physical Fitness, Physical Activity, BMI, and Sedentary Behavior? Results of HCSC Study

**DOI:** 10.3390/ijerph17031109

**Published:** 2020-02-10

**Authors:** Paweł Lisowski, Adam Kantanista, Michał Bronikowski

**Affiliations:** 1Department of School Practice, Faculty of Sport Science, 61-871 Poznań, Poland; 2Department of Physical Education and Lifelong Sports, Poznań University of Physical Education, 61-871 Poznań, Poland; kantanista@awf.poznan.pl; 3Department of Didactics of Physical Activity, Poznań University of Physical Education, 61-871 Poznań, Poland; bronikowski@awf.poznan.pl

**Keywords:** children, body status, motor competence, exercise

## Abstract

The transition from kindergarten to school is associated with a variety of negative changes. After entry to elementary school physical activity level decreases. Moreover, physical fitness level of children over the past decades have rapidly declined. Children are spending an increasing amount of time in the environments that require constant sitting. We evaluated the differences between boys and girls in physical fitness, frequency of undertaking of different forms of physical activity, prevalence of underweight and overweight, and time spent on sedentary behavior. A total of 212 first grade pupils (mean age 6.95 ± 0.43) from two standard urban schools in Poznań participated in the study. Compared to girls, boys obtained better results in 20-meter run (4.9 s and 5.0 s, *p* < 0.01), sit-ups (16.8 and 15.3, *p* < 0.05), six-minute run (829.7 m and 766.4 m, *p* < 0.001), and standing broad jump (106.8 cm and 99.7 cm, *p* < 0.01). In the sit-and-reach test girls achieved higher results than boys (17.0 cm and 14.4 cm, *p* < 0.001). There were no gender differences in prevalence of underweight and overweight. In conclusions, difference between genders should be taken into consideration during designing physical activity programs in the aspects of intensity and forms of physical activities.

## 1. Introduction

Physical fitness level during childhood has been identified as a predictor of current and future health status [1], but over the past decades it has rapidly declined [2]. Physical activity (PA) is related to many benefits and has potential to improve quality of life for school children [3]. However, after entering elementary school PA level decreases, both on weekdays and weekends [4] and it declines even further when reaching adolescence [5]. Lack of interest in PA may be due to the (un)motivational climate, and lack of support of parents and later of physical education (PE) teachers during a child’s first educational and sporting experiences. However, a school teaching routine with a great amount of time composed of sedentary activities does not motivate either [6]. Therefore, it seems important to realize what are the interests and levels of activity in those entering early education.

Physical fitness is an important factor for social development of children and is associated with a variety of health benefits [7]. There is consensus that being physically fit reduces the risk of cardiovascular diseases [8]. Studies also show that high level of physical fitness reduces overall and abdominal obesity, and can have a positive influence on mental and bone health [9], but it brings further benefits as well [10,11]. For example, Barnett et al. [12] reveals in a study on children and adolescents the cause-effect relationship of perceived sport competences and motor skills proficiency with the level of PA and fitness. Another study [13] shows that in children only pupils with lower physical fitness levels improved statistically significantly cardiorespiratory fitness. Chen et al. [14] found in Japanese children that unfavourable lifestyles in childhood are associated with poor quality of life in early adolescence and this has to be dealt with some organized educational approach to increasing level of PA in the early stages of education. It was academic performance that improved in the subsequent years for both boys and girls who increased their fitness level by more than 20 percentile points relative to the other students whose fitness did not change. Another longitudinal study [15], on a group of junior high school students indicated that both boys and girls who improved their physical fitness had better results at school in general.

In a study by Øygard [16] the 12-year follow-up research indicated that one of the most important factors associated with PA and fitness was their levels at baseline (especially in male pupils, in female ones it was education). Moreover, a study by Hussey [17] suggests that there are no significant sex differences in participation in light exercise, but significantly more boys than girls participated in at least 20 min hard exercise three or more times a week (53% boys, 28% girls). In a study of Jose et al. [18] it was noticed that for every minute taken to run 1.6 distance course in early school age it was 13% less likely for that child to stay persistently active in adulthood. In other words, the faster they ran, the better the chance they would stay active longer throughout their life. On the other hand, self-estimated level of sport skills and ‘fear of being judged or embarrassed by others’ are also reported barriers to regular PA together with lack of time [11].

In the first grade of primary school, children are subject to extensive physical changes [19]. During physical growth, speed, endurance and strength increase in both genders; however, development of muscular strength is slower than development of speed and endurance. Girls have better flexibility than boys at primary school age, and coordination based on individual range of performance [7,19]. Popovici et al. [20] reported that boys have a better lower limb and upper limb strength and abdominal muscle, and girls have a better balance compared to boys. In addition, boys at the age of seven to eight obtain considerable high endurance run degrees in comparison with girls. It is not clear whether gender differences, in motor task performance, exist in childhood, and moreover whether gender differences are established during development [21]. One needs to remember also that gender differences on motor performances and fitness are related to different trajectories of biological and environmental factors [22].

Recent studies have shown that PA is lower in girls than boys, and decreased cross-sectionally each year after age of five with corresponding increase in time spent sedentary [23]. These findings are likely to be due to the increase of school assignments and homework in children in the first grade [4]. Moreover, children are spending increasing amounts of time in environments that require sustained sitting—at school or at home. Owen et al. [24] noticed that schools and public spaces have been re-engineered in ways that minimize human movement and muscular activity so children move less and sit more. As a consequence, most of them do not meet the widely accepted World Health Organization [25] PA recommendations—at least 60 min every day of moderate-to-vigorous PA in order to benefit the health of children and adolescents [26]. The International Children’s Accelerometry Database (ICAD) has demonstrated that mean values for total volume of PA and moderate-to-vigorous PA both appear to decline steadily after the age of five years [23]. The study carried out in the UK suggests that PA is in decline in most individuals of both sexes by age six to seven years [27]. The start of primary school education is thus connected with decreasing of level of PA and moderate-to-vigorous PA.

Achieving PA recommendations is not protective against the health risks of sedentary behavior [6,24]. Mean percentage sedentary time increased with age, for both boys and girls [28]. Research of Hussey [17] showed that there is no sex difference in spending time in front of a screen, but boys are expending significantly more energy in regular activities than girls. Such gender differences in the intensity and choice of activities have been observed also in other studies [29]. Furthermore, some research indicated that children who had higher than average levels of PA, have also increased levels of sedentary behavior [30]. During the past twenty years, time for using computers, watching television and playing video games has dramatically increased; with nine in ten children using computers in school too much and for too long [31].

Insufficient level of PA and sedentary behavior are associated with excessive body mass. Childhood obesity is one of the most important challenges for the public health sector of the twenty-first century. In Europe, one in three boys and one in five girls aged six to nine years are overweight or obese. Prevalence of overweight varied from 18% to 29% in boys and from 18% to 28% in girls [32]. In Poland [33] 19.3% boys and 17.3% girls were overweight or obese. According to Tichá et al. [34] 2.4–2.8% of girls and 1.9–3.9% of boys aged seven years were underweight. Overweight may also be a problem that turns children off PA [35].

Farooq et al. [27] proposed that future research and public health policy should focus on preventing the decline in PA which begins in childhood, not adolescence, to provide an improved understanding of the determinants of the different PA trajectories. Research on young children’s physical fitness and PA are sparse [32] and the results of physical fitness are inconclusive [36]. Evidence on the association between sedentary and health among children needs more explanations [28] and relationship between spending leisure time and the development of physical fitness of children is relatively poorly represented [37]. However, despite the importance of this period of one’s life, few studies examined the changes in PA levels n five to seven years old children, and those that did examine such changes reported contrasting findings [38].

Comparison of health-related factors (physical fitness and activity) and factors risky for health (prevalence of underweight, sedentary behavior) between girls and boys are required to implement effective intervention programs. Therefore, the aims of the study were to evaluate differences between boys and girls in results of physical fitness tests, frequency of undertaking different forms of PA, prevalence of underweight, normal weight and overweight, and time spent on sedentary behavior on weekdays and weekends.

## 2. Materials and Methods

### 2.1. Participants

The study was conducted as a part of the international project “Healthy Children in Sound Communities” which has been implemented in European and also in some Asian countries (for details of that study look at Naul et al., 2012). In Poland the study included data collected from 212 first grade pupils (mean age 6.95 ± 0.43), 100 of whom were boys (body mass 26.38 ± 4.99 kg; height 125.7 ± 6.2 cm; body mass index (BMI) 16.59 ± 2.13 kg/m^2^), and 112 were girls (body mass 24.61 ± 4.61 kg; height 123.99 ± 5.47 cm; BMI 15.93 ± 2.23 kg/m^2^). The participants were recruited from the two standard urban schools in the city of Poznań. Written consent was obtained from the parents or guardians. Students were also informed about the anonymous and voluntary nature of their participation. The study protocol was approved by the Local Bioethics Committee of Karol Marcinkowski University of Medical Sciences in Poznań (decision No. 552/11).

### 2.2. Physical Fitness Measures

Physical fitness tests were performed during PE classes. Cardiorespiratory fitness was measured by a 6-min run around the perimeter of the rectangle with dimensions 9 × 18 m and the distance covered were registered. Strength endurance was evaluated by the number of sit-ups in 40 s. The purpose of the test was to measure the endurance of the abdominal and hip-flexor muscles. The test was performed on a mat with knees bent at right angles, and with feet flat on the floor and hooked underneath a gym ladder. The fingers were interlocked behind the head. The result was the maximum number of correctly performed sit ups in 40 s. Explosive power was assessed by standing broad jump test. A pupil performed a jump from a standing two feet on the ground position and behind the line for the longest distance in cm. Agility/speed was evaluated by the 20-m run involving running on the signal as fast as possible between the two lines 20 m apart. The sit and reach test was used for the assessment of flexibility of the lower back and hamstring muscles. The test involved sitting on the floor with legs out straight in front. Feet were placed with the soles flat against a box, shoulder-width apart. The subject’s task was to reach forward along the measuring line as far as possible. Excluding explosive power assessment (better results of two tests were taken)—all tests were performed once. All tests and procedure have been described earlier in details [39].

### 2.3. Evaluation of Prevalence of Underweight, Normal Body Weight and Overweight

Weight and height were measured with a Seca scale in light clothing (without shoes), while height was measured with an anthropometer to the nearest 0.5 cm. The body height and weight of participating girls and boys was determined and used to calculate body mass index (BMI). On the basis of this latter parameter, three groups with different BMI status were created: a) underweight, b) normal weight, and c) overweight. The qualification of participants into respective age- and gender-adjusted categories of body weight was based on the BMI cut-off values for children and adolescents proposed by Cole et al. [40] and Cole et al. [41].

### 2.4. Physical Activity and Sedentary Behavior Evaluation

To assess PA and sedentary behavior, a questionnaire used in the Healthy Children in Sound Communities (HCSC) project was used. The questionnaires were completed by parents of children participating in the study and took approximately 15 min to complete. To assess PA of pupils, their parents were asked: how often does your child undertake the following kinds of PA (for at least 20 min)? Possible answers were from “5–7 times a week” to “rare or never”. Parents could choose physical activities from the list or add a new one. Parents were also asked about sedentary behavior of their children: (1) “how long does your child watch TV/Videos/DVD each day”? and (2) “how long does your child play computer games/game console games each day”? Weekdays and weekends were evaluated. Results were analyzed in three categories: 0–1 h/day, between 1–2 h/day and more than 2 h/day.

### 2.5. Statistical Analysis

The t-test was used to evaluate gender differences in the results obtained during physical fitness test. For the comparison of the time spent on sedentary behavior and frequency of undertaking of different forms of PA according to gender, Mann–Whitney U test was used. The differences between genders in prevalence of underweight, normal weight and overweight were assessed based on percentage values using the test for difference between two independent sample proportions. Statistical significance was set at a probability of *p* < 0.05. Data were analyzed using STATISTICA software, version 13 (StatSoft Polska, Krakow, Poland).

## 3. Results

Differences between boys and girls in physical fitness tests are presented in Figure 1. Compared to girls, boys obtained better results in 20-m run (4.9 s and 5.0 s, *p* < 0.01), sit-ups (16.8 and 15.3, *p* < 0.05), 6-min run (829.7 m and 766.4 m, *p* < 0.001), and standing broad jump (106.8 cm and 99.7 cm, *p* < 0.01). In the sit-and-reach test girls achieved higher results than boys (17.0 cm and 14.4 cm, *p* < 0.001).

Frequency of undertaking of most popular PA forms by boys and girls are presented in Table 1. Boys more often than girls undertook biking (*p* < 0.05), running (*p* < 0.05), and different team games (*p* < 0.001). Girls more often than boys undertook roller skating (*p* < 0.05) and dancing (*p* < 0.001).

Body weight status of boys and girls is depicted in Figure 2. There were no differences between boys and girls in prevalence of underweight (boys: 8.0%, girls: 12.5%), normal body weight (boys: 68%, girls: 68.75%), and overweight (boys: 24.0%, girls: 18.75%).

Differences between boys and girls in time spent on sedentary behavior are presented in Figure 3. No difference was observed between boys and girls in time spent playing computer/console in weekdays. During weekends, boys spent more time playing computer/console games (*p* < 0.05); 23.2% of boys and 5.3% of girls spent more than two hour per day on this activity. During weekdays, boys spent more time than girls watching TV/DVD; 14.1% boys and 2.1% girls undertook this activity for more than two hours per day. There were no differences between boys and girls in time spent watching TV/DVD during weekends.

## 4. Discussion

In the study we evaluated differences between boys and girls in physical fitness, PA, body weight status and time spent on sedentary behavior. Except for the prevalence of underweight, normal body weight and overweight sex differences occurred in other examined variables.

We observed that 8% boys and 12.5% girls were underweight and 24.0% boys and 18.75% girls were overweight, but the sex differences were not significant. Prevalence of underweight boys is similar and prevalence of underweight girls is relatively higher in our study compared to the results obtained in the study of Tichá et al. [34] where International Obesity Task Force criteria linking adult cutoff points of overweight/obesity (25 kg/m^2^ and 30 kg/m^2^, respectively) to BMI centiles for children and adolescents for defining underweight, normal, and overweight were used. In Poland, a gradual increase in prevalence of overweight and obesity has been observed [42,43]. Additionally, we observed relatively lower percentage of normal body weight and higher percentage of overweight boys and girls in our than in Tichá et al. [34] study. However, the prevalence of overweight children was similar to the results of the study by Kułaga et al. [33]. We did not find differences in prevalence of underweight or overweight between boys and girls, but it is worth noting that almost 20% of overweight children enter the school education system and this is a relatively big ratio and may affect educational development. For example, in a study of Greenleaf and Weiller [44] it was found that PE teachers have lower expectations about the physical, cognition and social skills of overweight pupils. Children with overweight problems also engage in leisure-time PA less frequently and less regularly [45].

In terms of physical fitness, boys obtained better results than girls in all physical fitness components except flexibility. These results confirm previous research and show that girls have an advantage in flexibility [7,46], while boys perform better in conditioning components [7,46]. Although the participants were not in puberty period, differences in physical fitness between genders were significant. Boys appeared to prefer other sports or forms of PA than girls. Content of these activities including more endurance enhancing components may influence the development of the overall level of physical fitness. We found that boys compared to girls preferred more often biking, running, and team games, and less frequently roller skating and dance. These results partially confirm previous observations of Lampinen et al. [47], which found that the most common supervised form of PA among girls were dance and gymnastic and among boys’ ballgames. Additionally, the level of perceived sport competency (and its assessment by a PE teacher) has been found to be associated with involvement in PA and warrant of more regular engagement in adulthood [18]. Broad range and high movement literacy in fundamental sport-skills are factors determining children free-play activity involvement [48], but they are developed in preschool years with parents hopefully providing the most stimulating physical environment during the developmentally critical growth periods, thus creating opportunity to cultivate positive health behaviors [49]. The difference may be also the result of demographic and socio-cultural status. According to Seabra et al. [50] age, sex, socioeconomic status, sibling, parents/peers influence on PA of children and adolescents. It was indicated that girls participated in less intensive forms of PA, whereas boys preferred more vigorous activity [50]. There is also an issue of ‘fun factor’ influence on child’s PA. A study of Bremer and Cairney [51] reveals that it is not just the level of fundamental motor skills ability that plays a decisive role in child’s engagement in PA (along with physical fitness and body composition), but also participant’s enjoyment that makes the involvement sustainable. Children are less likely to remain engaged in PA in their later years if they do not enjoy the motor skill activities [51].

In our study, we indicated that boys spent more time playing computer/console games on weekends and more time watching TV/DVD on weekdays, compared to girls. These results confirm previous research [52] and show that boys tend to have more screen-based sedentary behavior, particularly using the computer, than girls. Lampinen et al. [47] noticed that time spent using computer, mobile phone, and playing mobile games was lower in girls than in boys, whereas time spent watching TV/DVD was similar in both genders. It is alarming that especially many boys spent more than 2 h/day on sedentary activities. Moreover, children spent on average almost half of waking time being sedentary [53], which is a habit considered to be a risk factor for obesity.

This study is not without limitations. Questionnaires on PA and sedentary behavior were completed by parents of children participated in the study. Reporting data by parents of children have some weaknesses, among others resulting from recall bias. Secondly, the relatively small sample size and sample only from one city could limit the generalizability of the findings. But it has provided a good data base for a PA intervention, well-tailored and designed to the needs and potential interest of the children aged six. For a better understanding of this age period and changes that children entering school go through, studies and developmental research in their design are needed in order to go beyond the current debate.

## 5. Conclusions

It was found that boys compared to girls obtained better results in physical fitness components excluding flexibility. Girls participated in less intensive forms of PA, whereas boys preferred more vigorous activity, which should become a pivotal point for education specialists in designing guidelines and curricula contents, but also assessment criteria and tools for the beginners to the education system—first graders. The difference between boys and girls should be taken into consideration during designing physical activity programs in aspects of intensity and forms of PA. In addition, taking into consideration the observation from our study that there were no sex differences in prevalence of underweight and overweight would be a value. However, in the study, it was found that that the percentage of boys and girls who were overweight was high, which should ring the bell in terms of considering introduction of preventing health education contents already in the early stages of schooling process. This may also help in reducing the time spend on sedentary activities and prevent excessive body weight gains at such early age, and help with forming the right patters (or help with reduction) of screening time, while increasing PA accordingly to the pupils’ interest and biological needs.

## Figures and Tables

**Figure 1 ijerph-17-01109-f001:**
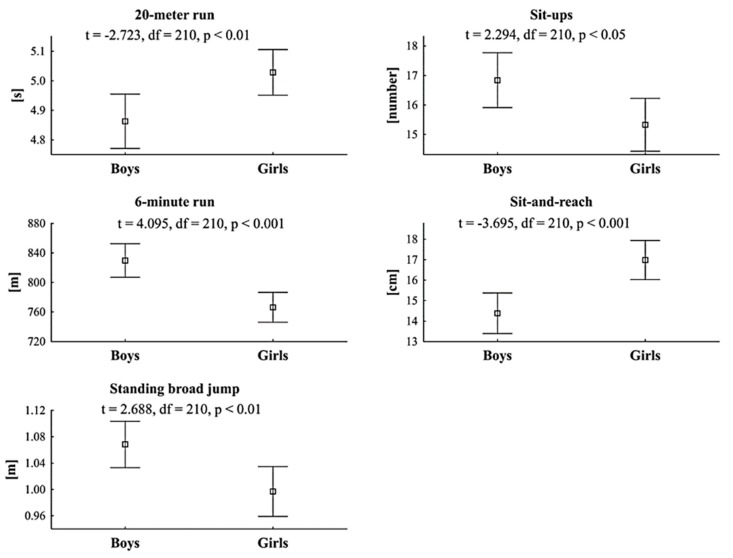
Differences between boys and girls in the results of physical fitness tests (boys, n = 100; girls, n = 112).

**Figure 2 ijerph-17-01109-f002:**
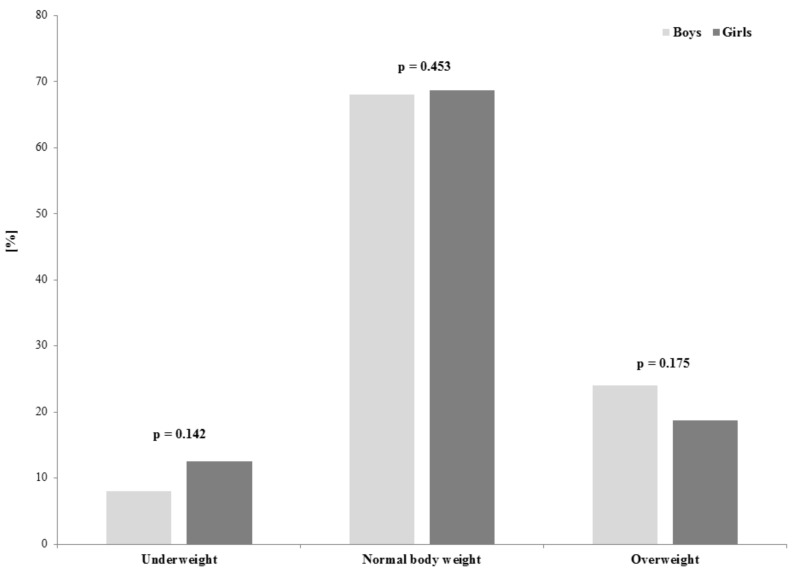
Differences in prevalence of underweight, normal body weight, and overweight in boys and girls.

**Figure 3 ijerph-17-01109-f003:**
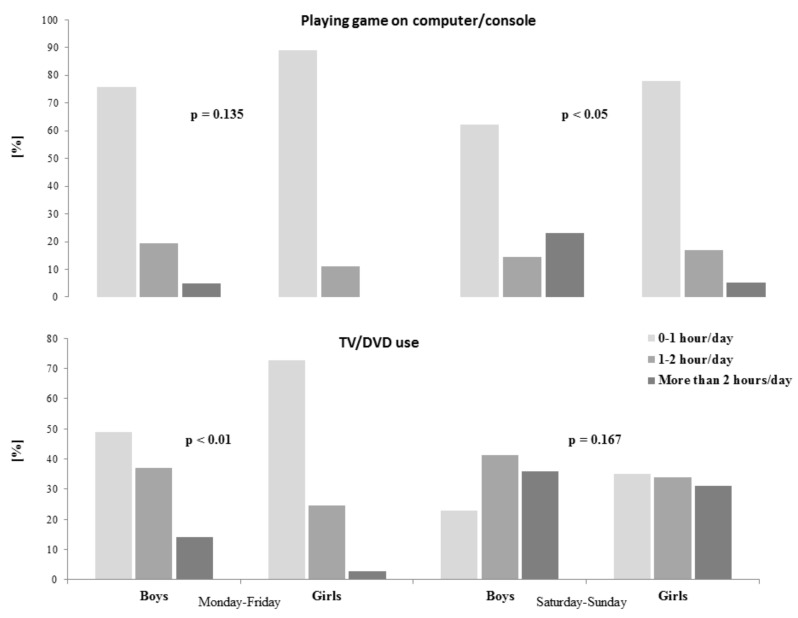
Differences between boys and girls in time spent on sedentary behavior.

**Table 1 ijerph-17-01109-t001:** Differences between boys and girls in frequency of undertaking of different forms of PA.

Forms of Physical Activity	Gender	Never or Rare	1–2 Times per Week	3–4 Times per Week	5–7 Times per Week	Z	P
Swimming	Boys, n = 85	40.0	57.6	1.2	1.2	0.275	0.783
	Girls, n = 91	45.1	48.4	4.4	2.2		
Biking	Boys, n = 90	17.8	40.0	30.0	12.2	2.451	<0.05
	Girls, n = 96	18.8	62.5	15.6	3.1		
Running	Boys, n = 81	38.3	32.1	9.9	19.8	2.268	<0.05
	Girls, n = 77	51.9	35.1	7.8	5.2		
Team games	Boys, n = 83	28.9	38.6	20.5	12.0	5.815	<0.001
	Girls, n = 75	76.0	22.7	0.0	1.3		
Roller skating	Boys, n = 74	75.7	18.9	5.4	0.0	−2.096	<0.05
	Girls, n = 88	58.0	27.3	12.5	2.3		
Skateboarding	Boys, n = 75	78.7	18.7	1.3	1.3	1.178	0.239
	Girls, n = 72	90.3	6.9	1.4	1.4		
Dancing	Boys, n = 73	58.9	41.1	0.0	0.0	−4.759	<0.001
	Girls, n = 88	22.7	59.1	11.4	6.8		

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
