# Peer review of "Are There Any Differences between First Grade Boys and Girls in Physical Fitness, Physical Activity, BMI, and Sedentary Behavior? Results of HCSC Study"

_ijerph, 2020, doi:10.3390/ijerph17031109_

Round 1

Reviewer 1 Report

In the abstract and in the first paragraph of the introduction you refer to the transition from kindergarten to elementary school. This is irrelevant to the study as you did not examine any transition-related data. Please eliminate these parts of the manuscript.

It may be better if you begin the abstract with the fourth sentence and eliminate the first three ones.

The first paragraph of the introduction is misleading. Please state more clearly the frame of the study. Eliminate the part about physical education as you did not examine anything about PE.

The second paragraph of the introduction, which is relevant to the study (i.e. children's fitness) should be expanded. You may put some more references regarding the importance of children's fitness for their health. Also describe in more detail, why being fit as a child may lead to better health as an adult.

The third paragraph of the introduction is relevant to the study. You may expand this somewhat by articulating the maturation level of boys and girls at the age of about 7 years old, (the age of participants of your study)

In the fifth paragraph, you mention research findings regarding gender differences in PA participation. You may move these to the fourth paragraph which outlines this theme.

In the fifth paragraph  you may employ the  findings to justify  your decision  to measure  both PA and sedentary behavior.

Methods

You don't need to provide the no  for the approval.

Physical  Fitness Measure

Rephrase to Physical Fitness Measures

In this section give more details about the tests. Do not assume that the readers are familiar with the test names. Your description of the Cardiorespiratory Fitness test is adequate. Do the same for the rest of the measures and provide references where possible.

Statistical analysis

This may be rephrased as results

The use of multiple t-tests enhances the possibility for Type – 2 error. You need to employ a Bonferonni adjustment for each subsequent t-test.

Discussion

Explain IOTF criteria

Author Response

Dear Reviewer 1

Thank you for your time and valuable comments and suggestions helping us to improve the quality of our paper.

In the abstract and in the first paragraph of the introduction you refer to the transition from kindergarten to elementary school. This is irrelevant to the study as you did not examine any transition-related data. Please eliminate these parts of the manuscript. It may be better if you begin the abstract with the fourth sentence and eliminate the first three ones.

This has been taken into consideration and corrected accordingly.

The first paragraph of the introduction is misleading. Please state more clearly the frame of the study. Eliminate the part about physical education as you did not examine anything about PE.

The reviewer was right – we did not examine any associations of PA with Physical Education and therefore linking the two in that particular part of the Introduction might have been misleading. We have delated that.

The second paragraph of the introduction, which is relevant to the study (i.e. children's fitness) should be expanded. You may put some more references regarding the importance of children's fitness for their health. Also describe in more detail, why being fit as a child may lead to better health as an adult.

Valid suggestion and we have added some relevant information (and references) to that part of the introduction.

The third paragraph of the introduction is relevant to the study. You may expand this somewhat by articulating the maturation level of boys and girls at the age of about 7 years old, (the age of participants of your study)

This has been expanded a little more and some references were also added to this part.

In the fifth paragraph, you mention research findings regarding gender differences in PA participation. You may move these to the fourth paragraph which outlines this theme.

In the fifth paragraph  you may employ the  findings to justify  your decision  to measure  both PA and sedentary behavior.

Some parts of those paragraphs have been relocated as suggested, some were cut off as having minor relation to the study rationale.

Methods

You don't need to provide the no  for the approval.

In our country it is needed when papers and studies are undergoing academic evaluation and therefore we will leave in the text.

Physical  Fitness Measure

Rephrase to Physical Fitness Measures

Thank you – it has been acknowledged and corrected.

In this section give more details about the tests. Do not assume that the readers are familiar with the test names. Your description of the Cardiorespiratory Fitness test is adequate. Do the same for the rest of the measures and provide references where possible.

This part has been extended and more details added to the description and also reference was provided for testing procedure in more detail.

Statistical analysis

This may be rephrased as results

The use of multiple t-tests enhances the possibility for Type – 2 error. You need to employ a Bonferonni adjustment for each subsequent t-test.

We didn't use Bonferroni adjustment because we compared boys and girls in terms of independent variables (fitness tests). Therefore, no data was analyzed in one model. We treated it as an independent comparison.

Discussion

Explain IOTF criteria

This has been added to the text.

Reviewer 2 Report

This study aimed to evaluate the physical fitness of first grade children, comparing the results by gender. The prevalence of body weight and habits to activity or sedentary behaviors of children have also been considered and described. Authors concluded that teachers/practitioners should carefully consider gender differences when designing PA programs.

The study addresses some current matters such as the PA design, the assessment of the fitness status and the assessment of the PA habits of young children, that lower as they enter in the school. The latter is very important matter of the physical literacy which aims to promote PA from childhood to achieve positive effects on health of the individuals and active lifestyles throughout the lifespan.

The study is well written and clearly describes the question research and findings. The introduction provides a wide report of the background and discussion properly supports the analysis of the results.

In my opinion, there are only two issues to be addressed:

Relationships among the prevalence of body weight, physical fitness and PA habits:

In the introduction and discussion, the authors properly considered literature from which overweight have been related to PA habits (e.g. lines 43-44, 198-199, or 230-231). In addition, reduced PA has been related to several barriers (e.g. lines 37-40, or 60-61), but I suppose overweight as well is among them (e.g. Stankov 2012 Overweight and obese adolescents: what turns them off physical activity? Int J Behav Nutr Phys Act. 2012 May 3;9:53).

Even if your study aimed to compare the results by gender, you classified children by prevalence in weight, and a noticeable number of overweight children has been highlighted. Therefore, I wonder whether some analysis or comment might be added and whether you can consider the physical fitness and PA habits also by “weight class”.  I don’t know whether your data are completely blind for privacy or whether you can somehow associate the weight class to the physical fitness and PA habits, to allow this. If yes, I think it would be very interesting and full of information to provide and discuss this additional analysis. After all, you linked several times overweight, obesity, sedentary, PA.

Conclusions:

I recognize the limitations you declared but, in my opinion, your conclusions ("to take into consideration the difference between gender") probably are reductive and simplify too much your results. I suggest to put more emphasis to the relevance of your investigation for PA assessment and promotion and to be more exhaustive about the issues to be addressed to differentiate by gender while designing PA programs, such as practical application/guidelines.

Minor issues:

Line 15: has instead of have

Line 125: was instead of were

Author Response

Dear Reviewer 2

Thank you for your suggestions and comments and your time that helped us improve the quality of the paper.

This study aimed to evaluate the physical fitness of first grade children, comparing the results by gender. The prevalence of body weight and habits to activity or sedentary behaviors of children have also been considered and described. Authors concluded that teachers/practitioners should carefully consider gender differences when designing PA programs.

This has been extended a little bit in the text and new references have been added.

The study addresses some current matters such as the PA design, the assessment of the fitness status and the assessment of the PA habits of young children, that lower as they enter in the school. The latter is very important matter of the physical literacy which aims to promote PA from childhood to achieve positive effects on health of the individuals and active lifestyles throughout the lifespan.

This issue has also been dealt with new references added to the text.

The study is well written and clearly describes the question research and findings. The introduction provides a wide report of the background and discussion properly supports the analysis of the results.

In my opinion, there are only two issues to be addressed:

Relationships among the prevalence of body weight, physical fitness and PA habits:

In the introduction and discussion, the authors properly considered literature from which overweight have been related to PA habits (e.g. lines 43-44, 198-199, or 230-231). In addition, reduced PA has been related to several barriers (e.g. lines 37-40, or 60-61), but I suppose overweight as well is among them (e.g. Stankov 2012 Overweight and obese adolescents: what turns them off physical activity? Int J Behav Nutr Phys Act. 2012 May 3;9:53).

Some new information and new references have been added concerning this issue, which the Reviewer rightly has indicated as very important.

Even if your study aimed to compare the results by gender, you classified children by prevalence in weight, and a noticeable number of overweight children has been highlighted. Therefore, I wonder whether some analysis or comment might be added and whether you can consider the physical fitness and PA habits also by “weight class”.  I don’t know whether your data are completely blind for privacy or whether you can somehow associate the weight class to the physical fitness and PA habits, to allow this. If yes, I think it would be very interesting and full of information to provide and discuss this additional analysis. After all, you linked several times overweight, obesity, sedentary, PA.

Unfortunately, We are unable to provide such information from the data that we have at this point as we aimed in our research to have a generic diagnosis of the state of all these factors as the base for further intervention programmes with first grade pupils, but this is a very good suggestion and we will certainly consider this in our further research studies and designs.

Conclusions:

I recognize the limitations you declared but, in my opinion, your conclusions ("to take into consideration the difference between gender") probably are reductive and simplify too much your results. I suggest to put more emphasis to the relevance of your investigation for PA assessment and promotion and to be more exhaustive about the issues to be addressed to differentiate by gender while designing PA programs, such as practical application/guidelines.

Thank you for this comment and we have adjusted the conclusions to be more exhaustive. 

Minor issues:

Line 15: has instead of have

This has been corrected.

Line 125: was instead of were

This has been corrected.

Round 2

Reviewer 1 Report

Thank you for addressing all the comments. The manuscript has improved significantly.

There are some spelling/syntax errors. Please check the manuscript thoroughly